# Comparison between Artificial Neural Network and Rigorous Mathematical Model in Simulation of Industrial Heavy Naphtha Reforming Process

Ali Al-Shathr [1], Zaidoon M. Shakor [1], Hasan Sh. Majdi [2], Adnan A. AbdulRazak [1,*] and Talib M. Albayati [1]

[1] Chemical Engineering Department, University of Technology-Iraq, Baghdad 10066, Iraq; Ali.R.MohammedJawad@uotechnology.edu.iq (A.A.-S.); Zaidoon.M.Shakor@uotechnology.edu.iq (Z.M.S.); Talib.M.Naieff@uotechnology.edu.iq (T.M.A.)
[2] Department of Chemical Engineering and Petroleum Industries, Al-Mustaqbal University College, Babylon 51001, Iraq; hasanshker1@gmail.com
* Correspondence: adnan.a.alsalim@uotechnology.edu.iq

**Abstract:** In this study, an artificial neural network (ANN) model was developed and compared with a rigorous mathematical model (RMM) to estimate the performance of an industrial heavy naphtha reforming process. The ANN model, represented by a multilayer feed forward neural network (MFFNN), had (36-10-10-10-34) topology, while the RMM involved solving 34 ordinary differential equations (ODEs) (32 mass balance, 1 heat balance and 1 momentum balance) to predict compositions, temperature, and pressure distributions within the reforming process. All computations and predictions were performed using MATLAB® software version 2015a. The ANN topology had minimum MSE when the number of hidden layers, number of neurons in the hidden layer, and the number of training epochs were 3, 10, and 100,000, respectively. Extensive error analysis between the experimental data and the predicted values were conducted using the following error functions: coefficient of determination ($R^2$), mean absolute error (MAE), mean relative error (MRE), and mean square error (MSE). The results revealed that the ANN ($R^2$ = 0.9403, MAE = 0.0062) simulated the industrial heavy naphtha reforming process slightly better than the rigorous mathematical model ($R^2$ = 0.9318, MAE = 0.007). Moreover, the computational time was obviously reduced from 120 s for the RMM to 18.3 s for the ANN. However, one disadvantage of the ANN model is that it cannot be used to predict the process performance in the internal points of reactors, while the RMM predicted the internal temperatures, pressures and weight fractions very well.

**Keywords:** heavy naphtha; reforming; mathematical model; artificial neural network; deactivation; catalyst

## 1. Introduction

In general, mathematical models can be categorized as deterministic or empirical; the deterministic models are constructed from first-principles equations, whereas empirical models are mathematical functions generalized to fit the data of one or more variables. Modeling catalytic refinery units is critical for designing, optimizing, and controlling tasks, but the mathematical modeling of these units poses several challenges, including the (1) assumptions used to simplify the models, (2) number of lumped components associated with the kinetic modeling, (3) complexity of the mathematical modeling, (4) catalyst activity decay with time, and (5) evaluation of physical properties under severe conditions [1,2].

Petroleum refineries consist of several thermal and catalytic units used to convert and separate petroleum fractions into useful products. Naphtha reforming units convert low-octane number heavy naphtha into a higher-octane number reformate that is the main feedstock for the blending unit to produce gasoline. Industrial naphtha reforming units include three or four reactors adiabatically operated at temperatures ranging from 450 to

520 °C and pressures ranging from 1 to 3.5 MPa [3,4]. Mathematical representations of these units can be done by integration of rigorous models that combine mass, heat, and momentum equations as well as kinetic models that govern the reactions of the hundreds of components within naphtha. In the past few years, several mathematical models have been developed to describe the behavior of naphtha reforming units. Tailleur (2012) used data obtained from commercial and micropilot plants to predict the reaction kinetics and catalyst deactivation parameters. The ordinary differential equations of mass and energy balance were solved using the Runge–Kutta–Fehlberg method in the axial and radial directions. [5] Iranshahi et al. (2014) developed mathematical and kinetic models to represent the continuous catalytic regeneration naphtha reforming process; the kinetic model consists of 32 pseudocomponents and 84 reactions. They obtained acceptable agreement between observed data and simulation results [6]. Elizalde and Ancheyta (2015) proposed the dynamic modeling of the catalytic naphtha reforming reactor using material and heat balances; the reaction network consisted of 20 components plus hydrogen and 53 chemical reactions [7]. Babaqi et al. (2018) simulated the continuous regeneration naphtha reforming process using a reaction network of 36 lumps and 55 reactions. The model has been validated by comparing its results with plant data, in which the mean relative error for the octane number, reformate yield, light gases, hydrogen yield, reactor temperature, and pressure was 1.3%, 2.5%, 0.93%, 0.43%, 1.03%, 2, and 0.6%, respectively [8]. Dong et al. (2018) described a continuous catalytic reforming process using a kinetic model of 27-lump, plug flow reactor model of a 4-zone parallel-series and an empirical catalyst deactivation model. The mean absolute prediction was found to be 0.76%, 0.42%, 0.90%, and 0.50% for paraffins, naphthenes, aromatics, and hydrogen, respectively [9]. Yusuf et al. (2019) used gPROMs® software for the steady-state and dynamic modeling of industrial catalytic reforming. The 3D representation of 25 profiles (i.e., concentration, temperature, research octane number [RON], and hydrogen yield) with respect to time and reactor height was estimated by solving the partial differential equations governing mass and heat transfer in the process [10]. Shakor et al. (2020) estimated the kinetic parameters for the kinetic model of 32 lumps and 132 reactions by fitting the model predictions with data obtained from the industrial heavy naphtha reforming process. They observed that after 1225 days the catalyst activity decayed to 58.8% of its original activity [11]. Studies concerning the modeling of catalytic heavy naphtha reforming vary greatly in the number of pseudocomponents in the reaction mixture and the number of reactions in the kinetic networks, and hence, in the model predictions, these models are also graded from moderate complexity to very complicated. Pishnamazi et al. (2020) developed a CFD-based simulation model to predict the amount of aromatic capacity, process efficiency, transfer rate, bed temperature, and pressure in case of changes in operating conditions for naphtha reforming units [12].

Ebrahimian and Iranshahi (2020) simulated the thermal coupling of naphtha reforming with propane ammoxidation using a one-dimensional homogenous model for two processes. A genetic algorithm (GA) was applied to optimize the operating conditions of the selected configuration (naphtha-series ammoxidation). The optimum temperature and feed flow rate and the number of tubes in three reactors were selected to be 776.94 K, 2086.2 kmol h$^{-1}$, and 395, respectively. [13] Yusuf et al. (2020) estimated the plant performance, temperature, and concentration profiles of the paraffins, naphthenes, and aromatics of semicatalyst regenerative commercial naphtha catalytic reforming using gPROMS software [14].

In a petroleum refinery, the operators need a fast, simple, and accurate methodology to estimate the process predictions. An artificial neural network (ANN) is one of the promising prediction methods that can be used to predict the performance of highly nonlinear operations. ANNs are widely used for the modeling and controlling of complex chemical processes, in which an ANN has been accurately used to simulate distillation columns, [15–17] heat exchangers, [18,19] and catalytic reactors [20,21]. A survey of the literature found that very few studies have investigated the application of ANNs in the

modeling of heavy naphtha reforming. Sadighi and Mohaddecy (2013) developed a layered-recurrent artificial neural network to simulate an industrial fixed-bed catalytic reforming unit. They observed that the ANN could simulate the research octane number (RON), flow rate of produced gasoline, and octane barrel level with a mean absolute deviation of 0.238%, 0.813%, and 0.853%, respectively [22]. Elfghi (2016) compared the response surface methodology (RSM) and artificial neural network (ANN) for modeling catalytic naphtha reforming units and optimized the RON of a produced gasoline. The ANN methodology showed a very obvious advantage over RSM. A maximum RON of 98 was obtained at the optimal conditions (T = 521 °C, P = 3.76 MPa, LHSV = 2.02 h$^{-1}$) [23].

The objective of this work was to explore the use of two different models to predict the performance of the heavy naphtha reforming process. The rigorous mathematical model (RMM) was selected as the deterministic model, while the artificial neural network (ANN) was selected as the empirical model.

## 2. Process Description and Data Collection

The flow sheet of the semiregenerative heavy naphtha reforming unit which is located in the Al-Doura Refinery (Baghdad, Iraq) is shown in Figure 1. This unit consists of four reactors with four interstage heaters in a series and containing a Pt/Al$_2$O$_3$ catalyst. The reactors' inlet temperature was 470 °C, feed pressure was 2.75 MPa, and hydrogen-to-hydrocarbons feed ratio was 7 mol/mol. Gas chromatography/mass spectrometry (GC/MS) was used to analyze the samples collected from the feed and reactors effluent.

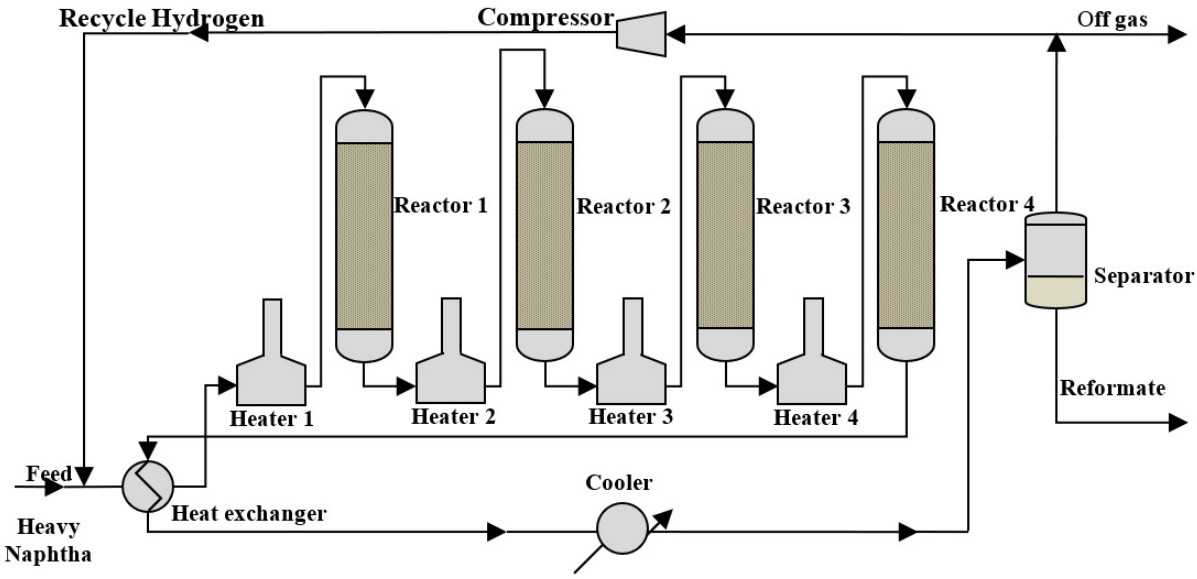

**Figure 1.** Flow sheet of the semi-regenerative naphtha reforming unit.

Ten datasets were collected from the commercial heavy naphtha reforming process in a long time period (1225 days), starting after using a fresh catalyst in the reactors. Each dataset contains information about the weight fractions of the streams, catalyst loading, and operating conditions for the commercial heavy naphtha reforming unit (see Table 1).

**Table 1.** Dataset of the weight fractions of the streams, catalyst loading, and operating conditions.

| Lump | Feed | Reactor A | Reactor B | Reactor C | Reactor D |
|---|---|---|---|---|---|
| | | | **Weight Fraction** | | |
| P1 | 0 | 0 | 0 | 0 | 0 |
| P2 | 0 | 0 | 0 | 0 | 0 |
| P3 | 0.72 | 0 | 0 | 0 | 0 |
| $n$-P4 | 0.43 | 0.22 | 0.32 | 0.31 | 0.21 |
| $n$-P5 | 0.52 | 0.81 | 0.86 | 0.79 | 0.63 |
| $n$-P6 | 3.67 | 2.38 | 2.37 | 2.22 | 1.68 |
| $n$-P7 | 0 | 3.53 | 3.28 | 2.63 | 1.89 |
| $n$-P8 | 8.87 | 5.57 | 3.2 | 2.17 | 1.58 |
| $n$-P9 | 6.58 | 0.26 | 0.25 | 0.25 | 0.48 |
| $n$-P10 | 2.42 | 3.28 | 1.88 | 0.22 | 0.17 |
| $n$-P11 | 0.2 | 0.25 | 0.33 | 0.36 | 0.11 |
| $i$-P4 | 0.35 | 0.15 | 0.32 | 0.15 | 0.18 |
| $i$-P5 | 0.36 | 0.94 | 0.93 | 0.79 | 0.75 |
| $i$-P6 | 3.17 | 5.19 | 6.12 | 6.07 | 5.16 |
| $i$-P7 | 4.57 | 6.05 | 6.43 | 6.05 | 5.11 |
| $i$-P8 | 9.77 | 7.11 | 5.9 | 4.92 | 3.72 |
| $i$-P9 | 13.28 | 10.02 | 7.85 | 5.93 | 3.71 |
| $i$-P10 | 9.02 | 9.15 | 7.46 | 7.14 | 3.02 |
| $i$-P11 | 1.19 | 2.8 | 3.22 | 4.77 | 0.36 |
| MCP | 0.26 | 0.23 | 0.28 | 0.28 | 0.27 |
| N6 | 2.07 | 0 | 0 | 0 | 0 |
| N7 | 4.84 | 0.57 | 0.45 | 0.39 | 0.37 |
| N8 | 7.02 | 0.86 | 0.81 | 0.68 | 0.49 |
| N9 | 0.95 | 0 | 0 | 0 | 0 |
| N10 | 3.9 | 0 | 0 | 0 | 0 |
| N11 | 0 | 0 | 0 | 0 | 0 |
| A6 | 0.35 | 0.8 | 1.15 | 1.35 | 1.32 |
| A7 | 2.84 | 6.33 | 8.86 | 10.95 | 12.27 |
| A8 | 8.31 | 14.5 | 19.14 | 23.53 | 26.94 |
| A9 | 3.97 | 19 | 18.59 | 18.05 | 29.58 |
| A10 | 0.37 | 0 | 0 | 0 | 0 |
| A11 | 0 | 0 | 0 | 0 | 0 |
| Catalyst loading (kg) | | 2700 | 4500 | 4750 | 5875 |
| Feed temperature (°C) | | 470 | 470 | 475 | 475 |
| Outlet temperature (°C) | | 431 | 454 | 476 | 474 |
| Liquid feed flow rate (m$^3$/hr) | 33.5 | | | | |
| Inlet pressure to reactor A (MPa) | 3.04 | | | | |
| Outlet pressure from reactor D (MPa) | 2.25 | | | | |

## 3. Modeling

### 3.1. Rigorous Mathematical Model

To simplify the heavy naphtha reforming process, the following simplification assumptions were considered: (1) Plug flow in reactors. (2) Negligible mass and energy transfer in radial direction [24]. (3) First-order homogeneous gas phase reactions [1,2,25]. (4) Steady state adiabatic reactors. (5) Negligible heat loss to atmosphere. The physical and thermodynamic properties of the pseudocomponents was assumed to be equal to the properties of the main component for these pseudocomponents. The physical properties were obtained from reference [26].

Mass and energy changes with respect to catalyst weight are represented in ordinary differential Equations (1) and (2), respectively [27,28]:

$$\frac{dF_i}{dw} = -\sum_{j}^{m} S_{i,j} r_j \tag{1}$$

$$\frac{dT}{dw} = \frac{-\sum_{j=1}^{m} r_j \Delta H_{Rj}}{\sum_{i=1}^{n} F_i C p_i} \tag{2}$$

The reaction rate was calculated by Equation (3), and a modified Arrhenius Equation (4) was used to calculate reaction rate constant.

$$r_j = k_j P^{\,o}_{\,i} \tag{3}$$

$$k_j = k_j^o exp\left[\frac{E_{Aj}}{R}\left(\frac{1}{T_o} - \frac{1}{T}\right)\right]\left(\frac{p_t}{p_{to}}\right)^\alpha \tag{4}$$

where: $n$ represents the number of components, $m$ is the reactions number.

The specific heat of components was estimated according to third order polynomial [28]:

$$Cp_i = A_i + B_i T + C_i T^2 + D_i T^3 \tag{5}$$

Heat of reaction is calculated by:

$$\Delta H_r^o = \sum \Delta H_{r,prodcts}^o - \sum \Delta H_{r,reactants}^o \tag{6}$$

$$\Delta H_t = \Delta H_r^o + \int_{T_o}^T \left[\sum \Delta Cp_{prodcts} - \sum \Delta Cp_{reactants}\right] dT \tag{7}$$

The Ergun equation was used to predict the pressure drop [29]:

$$\frac{dP_t}{dz} = -\frac{G}{\rho g_c D_p}\left(\frac{1-\varepsilon_b}{\varepsilon_b^3}\right)\left(150\frac{(1-\varepsilon_b)\mu}{D_p} + 1.75G\right) \tag{8}$$

Equation (9) was used to convert the difference in the reactor length to the difference in the catalyst weight.

$$\frac{dw}{dz} = A_C \rho_C (1 - \varepsilon_b) \tag{9}$$

Time dependent catalyst deactivation was used to represent the rate of catalyst decay with time as shown in Equation (10) [30]:

$$a = exp(-k_d t) \tag{10}$$

The proposed kinetic model involves 32 pseudo-components and 132 reactions; the pseudo-components are 1 to 11 carbon atoms of normal paraffins, 4 to 11 carbon atoms of isoparaffins, 6 to 11 carbon atoms of naphthenes, and 6 to 11 carbon atoms of aromatics. The kinetic model consists of isomerization, hydrocracking, dehydrogenation, dehydrocyclization, and hydrodealkylation reactions. The activation energies were grouped into nine activation energies according to reaction type. The estimated pre-exponential factors and activation energies are presented in Tables 2 and 3 respectively.

### 3.2. Artificial Neural Network Model

Artificial Neural networks are a series of mathematical algorithms that mimic the processes of the human brain to estimate relationships between massive amounts of data. Based on a set of experimental data, ANN models create nonlinear relationships that relate the independent and dependent variables. A multilayer feed forward neural network is the most commonly used topology in neural networks [31]. The connections between the input and hidden layer are calculated according to values called weights, which represent the strength of the connection between neurons. The output of the jth and kth nodes of the hidden layers are given by the following equations [29]:

$$net_j = \sum_{i=1}^I W_{i,j} X_i$$
$$output_j = f(net_j) \tag{11}$$

$$net_k = \sum_{n=1}^{N} W_{n,k} X_n$$
$$\text{output}_k = f(net_k) \tag{12}$$

**Table 2.** Pre-Exponential factors of predicted kinetic model.

| Reaction Step | $k_o$ | Reaction Step | $k_o$ | Reaction Step | $k_o$ | Reaction Step | $k_o$ |
|---|---|---|---|---|---|---|---|
| $nP_4 \rightarrow iP_4$ | 0.0034 | $nP_{10} \rightarrow N_{10} + H_2$ | 0.0074 | $iP_9 + H_2 \rightarrow iP_5 + nP_4$ | 0.0002 | $N_9 + H_2 \rightarrow N_7 + P_2$ | 0.0011 |
| $nP_5 + H_2 \rightarrow nP_4 + P_1$ | 0.0002 | $nP_{11} + H_2 \rightarrow nP_{10} + P_1$ | 0.0010 | $iP_9 \rightarrow N_9 + H_2$ | 0.3145 | $N_9 \rightarrow A_9 + 3H_2 .$ | 0.8036 |
| $nP_5 + H_2 \rightarrow nP_3 + P_2$ | 0.0111 | $nP_{11} + H_2 \rightarrow nP_9 + P_2$ | 0.0007 | $iP_{10} \rightarrow nP_{10}$ | 0.0155 | $N_{10} + H_2 \rightarrow nP_{10}$ | 0.0317 |
| $nP_5 \rightarrow iP_5$ | 0.0073 | $nP_{11} + H_2 \rightarrow nP_8 + P_3$ | 0.0010 | $iP_{10} + H_2 \rightarrow iP_9 + P_1$ | 0.0012 | $N_{10} + H_2 \rightarrow iP_{10} .$ | 0.1057 |
| $nP_6 + H_2 \rightarrow nP_5 + P_1$ | 0.0053 | $nP_{11} + H_2 \rightarrow nP_7 + nP_4$ | 0.0005 | $iP_{10} + H_2 \rightarrow iP_8 + P_2$ | 0.0021 | $N_{10} + H_2 \rightarrow N_9 + P_1$ | 0.0058 |
| $nP_6 + H_2 \rightarrow nP_4 + P_2$ | 0.0001 | $nP_{11} + H_2 \rightarrow nP_6 + nP_5$ | 0.0012 | $iP_{10} + H_2 \rightarrow iP_7 + P_3$ | 0.0011 | $N_{10} + H_2 \rightarrow N_8 + P_2$ | 0.0083 |
| $nP_6 + H_2 \rightarrow 2P_3$ | 0.0013 | $nP_{11} \rightarrow iP_{11}$ | 0.0019 | $iP_{10} + H_2 \rightarrow iP_6 + nP_4$ | 0.0000 | $N_{10} + H_2 \rightarrow N_7 + P_3$ | 0.0041 |
| $nP_6 \rightarrow iP_6$ | 0.0763 | $nP_{11} \rightarrow N_{11} + H_2$ | 0.0024 | $iP_{10} + H_2 \rightarrow iP_5 + nP_5$ | 0.0018 | $N_{10} \rightarrow A_{10} + 3H_2$ | 7.5196 |
| $nP_6 \rightarrow MCP + H_2$ | 0.0037 | $iP_4 \rightarrow nP_4$ | 0.0031 | $iP_{10} \rightarrow N_{10} + H_2$ | 0.0039 | $N_{11} + H_2 \rightarrow nP_{11}$ | 0.1008 |
| $nP_6 \rightarrow N_6 + H_2$ | 0.0016 | $iP_5 \rightarrow nP_5$ | 0.0162 | $iP_{11} \rightarrow nP_{11}$ | 0.0296 | $N_{11} + H_2 \rightarrow iP_{11}$ | 1.4912 |
| $nP_7 + H_2 \rightarrow nP_6 + P_1$ | 0.0014 | $iP_5 + H_2 \rightarrow iP_4 + P_1$ | 0.0003 | $iP_{11} + H_2 \rightarrow iP_{10} + P_1$ | 0.0072 | $N_{11} + H_2 \rightarrow N_{10} + P_1$ | 0.0018 |
| $nP_7 + H_2 \rightarrow nP_5 + P_2$ | 0.0006 | $iP_5 + H_2 \rightarrow P_3 + P_2$ | 0.0126 | $iP_{11} + H_2 \rightarrow iP_9 + P_2$ | 0.3902 | $N_{11} + H_2 \rightarrow N_9 + P_2$ | 0.0025 |
| $nP_7 + H_2 \rightarrow nP_4 + P_3$ | 0.0000 | $iP_6 \rightarrow nP_6$ | 0.0261 | $iP_{11} + H_2 \rightarrow iP_8 + P_3$ | 0.0021 | $N_{11} + H_2 \rightarrow N_8 + P_3$ | 0.0011 |
| $nP_7 \rightarrow iP_7$ | 0.0670 | $iP_6 + H_2 \rightarrow iP_5 + P_1$ | 0.0021 | $iP_{11} + H_2 \rightarrow iP_7 + nP_4$ | 0.0003 | $N_{11} \rightarrow A_{11} + 3H_2$ | 0.0009 |
| $nP_7 \rightarrow N_7 + H_2$ | 0.0412 | $iP_6 + H_2 \rightarrow iP_4 + P_2$ | 0.0001 | $iP_{11} + H_2 \rightarrow iP_6 + nP_5$ | 0.0007 | $A_6 + 3H_2 \rightarrow N_6$ | 0.0137 |
| $nP_8 + H_2 \rightarrow nP_7 + P_1$ | 0.0003 | $iP_6 + H_2 \rightarrow 2P_3$ | 0.0009 | $iP_{11} \rightarrow N_{11} + H_2$ | 0.0248 | $A_7 + 4H_2 \rightarrow nP_7$ | 0.0126 |
| $nP_8 + H_2 \rightarrow nP_6 + P_2$ | 0.0011 | $iP_6 \rightarrow MCP + H_2$ | 0.0037 | $MCP + H_2 \rightarrow nP_6$ | 0.0230 | $A_7 + 4H_2 \rightarrow iP_7$ | 0.0152 |
| $nP_8 + H_2 \rightarrow nP_5 + P_3$ | 0.0006 | $iP_6 \rightarrow N_6 + H_2$ | 0.0005 | $MCP + H_2 \rightarrow iP_6$ | 0.0994 | $A_8 + 4H_2 \rightarrow nP_8$ | 0.0029 |
| $nP_8 + H_2 \rightarrow 2nP_4$ | 0.0001 | $iP_7 \rightarrow nP_7$ | 0.0351 | $MCP \rightarrow N_6$ | 0.0097 | $A_8 + 4H_2 \rightarrow iP_8$ | 0.0098 |
| $nP_8 \rightarrow iP_8$ | 0.1052 | $iP_7 + H_2 \rightarrow iP_6 + P_1$ | 0.0099 | $N_6 + H_2 \rightarrow nP_6$ | 0.0035 | $A_8 + H_2 \rightarrow A_7 + P_1$ | 0.0010 |
| $nP_8 \rightarrow N_8 + H_2$ | 0.0116 | $iP_7 + H_2 \rightarrow iP_5 + P_2$ | 0.0045 | $N_6 + H_2 \rightarrow iP_6$ | 0.4533 | $A_9 + 4H_2 \rightarrow nP_9$ | 0.0165 |
| $nP_9 + H_2 \rightarrow nP_8 + P_1$ | 0.0017 | $iP_7 + H_2 \rightarrow iP_4 + P_3$ | 0.0001 | $N_6 \rightarrow MCP$ | 0.0100 | $A_9 + 4H_2 \rightarrow iP_9$ | 0.0508 |
| $nP_9 + H_2 \rightarrow nP_7 + P_2$ | 0.0023 | $iP_7 \rightarrow N_7 + H_2$ | 0.0155 | $N_6 \rightarrow A_6 + 3H_2$ | 0.6178 | $A_9 + H_2 \rightarrow A_8 + P_1$ | 0.0033 |
| $nP_9 + H_2 \rightarrow nP_6 + P_3$ | 0.0006 | $iP_8 \rightarrow nP_8$ | 0.0344 | $N_7 + H_2 \rightarrow nP_7$ | 0.0016 | $A_9 + H_2 \rightarrow A_7 + P_2$ | 0.0028 |
| $nP_9 + H_2 \rightarrow nP_5 + nP_4$ | 0.0001 | $iP_8 + H_2 \rightarrow iP_7 + P_1$ | 0.0037 | $N_7 + H_2 \rightarrow iP_7$ | 0.0086 | $A_{10} + 4H_2 \rightarrow nP_{10}$ | 0.0033 |
| $nP_9 \rightarrow iP_9$ | 0.4088 | $iP_8 + H_2 \rightarrow iP_6 + P_2$ | 0.0044 | $N_7 \rightarrow A_7 + 3H_2$ | 0.7168 | $A_{10} + 4H_2 \rightarrow iP_{10}$ | 0.7510 |
| $nP_9 \rightarrow N_9 + H_2$ | 0.0070 | $iP_8 + H_2 \rightarrow iP_5 + P_3$ | 0.0003 | $N_8 + H_2 \rightarrow nP_8$ | 0.0373 | $A_{10} + H_2 \rightarrow A_9 + P_1$ | 0.9012 |
| $nP_{10} + H_2 \rightarrow nP_9 + P_1$ | 0.4301 | $iP_8 + H_2 \rightarrow nP_4 + iP_4$ | 0.0001 | $N_8 + H_2 \rightarrow iP_8$ | 0.1945 | $A_{10} + H_2 \rightarrow A_8 + P_2$ | 0.0038 |
| $nP_{10} + H_2 \rightarrow nP_8 + nP_2$ | 0.0331 | $iP_8 \rightarrow N_8 + H_2$ | 0.0861 | $N_8 + H_2 \rightarrow N_7 + P_1$ | 0.0013 | $A_{10} + H_2 \rightarrow A_7 + P_3$ | 0.0008 |
| $nP_{10} + H_2 \rightarrow nP_7 + nP_3$ | 0.0035 | $iP_9 \rightarrow nP_9$ | 0.0149 | $N_8 \rightarrow A_8 + 3H_2$ | 0.5233 | $A_{11} + 4H_2 \rightarrow nP_{11}$ | 0.0179 |
| $nP_{10} + H_2 \rightarrow nP_6 + nP_4$ | 0.0004 | $iP_9 + H_2 \rightarrow iP_8 + P_1$ | 0.0047 | $N_9 + H_2 \rightarrow nP_9$ | 0.0020 | $A_{11} + 4H_2 \rightarrow iP_{11}$ | 0.0501 |
| $nP_{10} + H_2 \rightarrow 2nP_5$ | 0.0052 | $iP_9 + H_2 \rightarrow iP_7 + P_2$ | 0.0004 | $N_9 + H_2 \rightarrow iP_9$ | 0.0182 | $A_{11} + H_2 \rightarrow A_{10} + P_1$ | 0.0038 |
| $nP_{10} \rightarrow iP_{10}$ | 0.0592 | $iP_9 + H_2 \rightarrow iP_6 + P_3$ | 0.0008 | $N_9 + H_2 \rightarrow N_8 + P_1$ | 0.0026 | $A_{11} + H_2 \rightarrow A_9 + P_2$ | 0.0435 |

**Table 3.** Activation energies of predicted kinetic model.

| | $E_A \left( \frac{J}{mol} \right)$ | $\alpha$ |
|---|---|---|
| Dehydrocyclization of Paraffin's $(P_n \rightarrow N_n)$ | 52,712 | −0.66 |
| Hydrocracking of Paraffin's $(P_n \rightarrow P_{n-1} + P_i)$ | 72,254 | 0.20 |
| Isomerization of Paraffin's $(iP_n \longleftrightarrow nP_n)$ | 135,455 | 0.00 |
| Dehydrogenation of Naphthenes $(N_n \rightarrow A_n)$ | 40,528 | 0.11 |
| Hydrodealkylation of Naphthenes $(N_n \rightarrow N_{n-1} + P_i)$ | 186,470 | 0.49 |
| Ring Opening of Naphthenes $(N_n \rightarrow P_n)$ | 23,345 | 0.96 |
| Hydrodealkylation of Aromatics $(A_n \rightarrow A_{n-1} + P_i)$ | 138,888 | 0.29 |
| Ring Opening of Aromatics $(A_n \rightarrow P_n)$ | 138,635 | 1.17 |
| Hydrogenation of Aromatics $(A_n \rightarrow N_n)$ | 149,622 | 0.70 |

The transfer functions (sigmoid) occur between the layers of the ANN and relate the input and output of the transfer function [32]. Three transfer functions (i.e., tangent, logarithmic, and linear) are most widely employed in neural network models [33]. As with the formation of the output of the hidden nodes, the ANN outputs are produced based on the summation of incoming weighted signals of hidden nodes passing through a specific transfer function $f_o$.

$$Y_j = f_o \left( \sum_{i=1}^{n} W_{I,j} X_i + b \right) \tag{13}$$

Designing an Artificial Neural Network

The ANN model was designed to predict the performance of catalytic heavy naphtha using a set of input parameters. A multilayered feed forward neural network (MFFNN)

was used in this study. The input to the ANN model contained 36 nodes, including feed weight fractions, inlet temperature, inlet pressure, catalyst weight, and dataset time, while the output of the model contained 34 nodes, including product weight fractions, outlet temperature, and outlet pressure. There is a huge difference in the ranges of the ANN input-output data (shown in Table 4), where the weight fractions are within the range of 0–1, while the other variables are much greater. The temperatures, pressures, catalyst weights, and times were normalized within the range from 0–1 according to the Equation (14) [34]:

$$x_i^{norm} = \frac{x_i - x_{min}}{x_{max} - x_{min}} \tag{14}$$

**Table 4.** Range of ANN model variables.

| Variable | Minimum | Maximum |
|---|---|---|
| Mole fraction | 0 | 0.2353 |
| Temperature (°K) | 697.1500 | 743.1500 |
| Pressure (MPa) | 2.6700 | 3.0400 |
| Catalyst weight (kg) | 2700 | 5875 |
| Time (day) | 0 | 1225 |

The weight fractions of some components cannot be normalized using Equation (14) because its values may be equal to zero, making the value of the normalized variable equal to infinity as a result of division by zero.

The predicted data of the temperature and pressure were denormalized to their regional values using Equation (15):

$$x_i = x_{min} + (x_{max} - x_{min}) \times x_{i,Pred} \tag{15}$$

The tangent sigmoid transfer function for the hidden layer *f(X)* is given by Equation (16):

$$f(X) = \frac{2}{(1 + e^{-2n})} - 1 = tan\ sig(X) \tag{16}$$

Different topologies with one, two, and three hidden layer(s) and varying numbers of neurons (1 to 34) in hidden layer(s) were iteratively tested to achieve the optimum topology. The mean square error (MSE) was used to quantify the performance of the neural network topology according to Equation (17):

$$MSE = \frac{1}{N} \sum_{i=1}^{N} \left( y_{exp,i} - y_{pred,i} \right)^2 \tag{17}$$

The averages of the MSE values for all neural network topologies are presented in Figure 2. When using only one hidden layer, the levels of the MSE were higher than when using two or three hidden layers (Figure 2a). In addition, the unacceptable values of the MSE were estimated in different values of neurons. Using two hidden layers of an equal number of neurons produced lower values of the MSE than when using only one hidden layer, especially for neurons higher than 9 (Figure 2b). As shown in Figure 2c, acceptable values of MSE were estimated using three hidden layers of equal numbers of neurons, especially when the number of neurons exceeded 7, in which case the values of the MSE were stable for the lowest level. Therefore, three hidden layers, each comprised of 10 neurons, were selected as the optimum ANN topology to represent the results of the heavy naphtha reforming process.

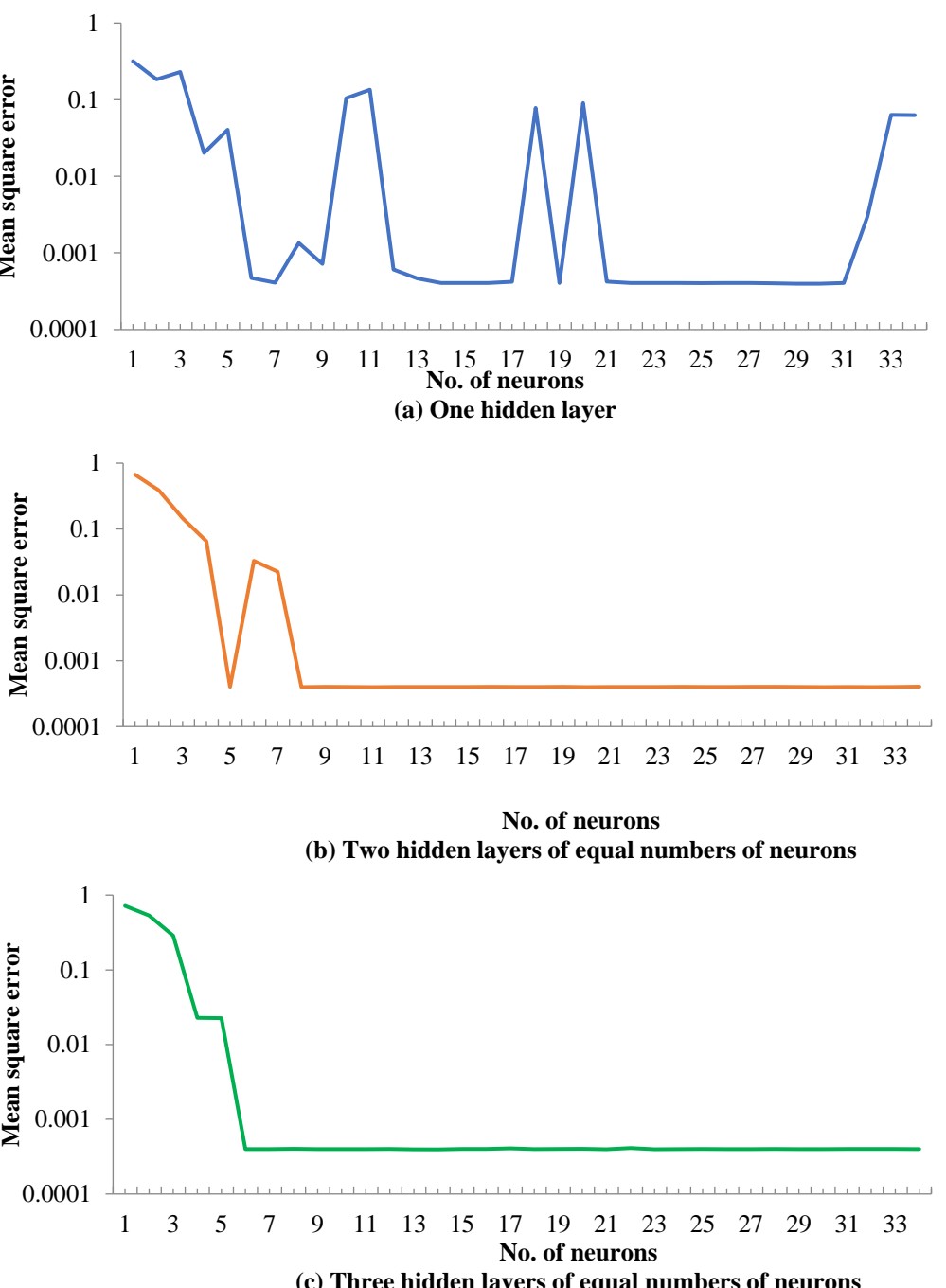

**Figure 2.** Mean square error achieved for (**a**) one hidden layer, (**b**) two hidden layers and (**c**) three hidden layers.

The optimum neural network topology (36–10–10–10–34) is plotted as shown in Figure 3, this topology having 36 neurons in the input layer, 10 neurons for each one of three hidden layers, and 34 neurons in the output layer.

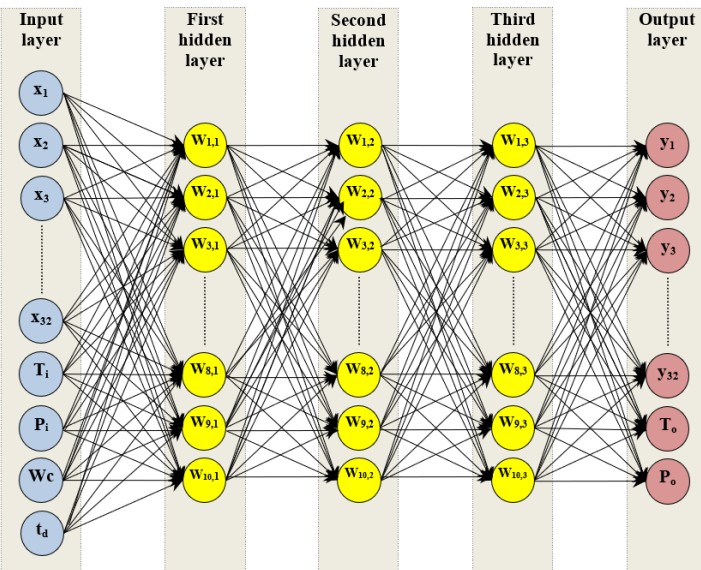

**Figure 3.** Optimum topology of artificial neural network model.

Figure 4 illustrates a graph of the MSE versus the epochs number using the optimum neural network topology. The epochs number reveals the number of iterations until reaching the optimal value of the objective function (minimum MSE). Obviously, the MSE decreases with an increase in the number of epochs. The minimum value of the MSE (0.00004) was observed when the epoch number reached 99,951, while further increase in the epoch number did not have a noticeable effect of the value of the MSE. Therefore, the maximum number of epochs was selected as 100,000 during ANN modeling. The parameters of the ANN model are summarized in Table 5.

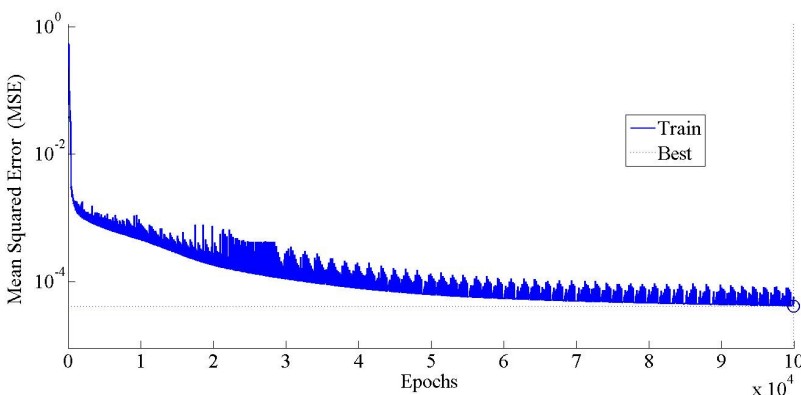

**Figure 4.** Predicted MSE with respect to the number of epochs.

**Table 5.** Parameters of the artificial neural network model.

| Input Layer | Input Data (36 Features) |
|---|---|
| Number of hidden layers | 3 |
| Hidden neuron for each hidden layer | 10 |
| Output layer | Prediction of naphtha reforming performance (34) |
| Performance function | Mean squared error (MSE) |
| Activation function | Sigmoid |
| Learning rate | 0.0001 |
| Maximum number of iterations | 100,000 |
| Gradient | 0.00001 |
| Type of activation sigmoid | Tan-Sigmoid |
| Algorithm used for training | Levenberg–Marquardt |

## 4. Statistical Analyses

Four statistical criteria were estimated to evaluate the performance of the two models developed in this study: correlation coefficient ($R^2$), mean absolute error (MAE), mean relative error (MRE), and mean squared error (MSE). The coefficient of determination ($R^2$) was calculated using Equation (18) [35]:

$$R^2 = 1 - \frac{\sum_{i=1}^{N} \left( y_{exp,i} - y_{pred,i} \right)^2}{\sum_{i=1}^{N} \left( y_{exp,i} - \overline{y}_{exp} \right)^2} \tag{18}$$

The mean absolute error (MAE) and mean relative error (MRE) were calculated using Equations (19) and (20), respectively.

$$MAE = \frac{1}{N \times M} \sum_{j=1}^{M} \sum_{i=1}^{N} \left| y_{i,j}^{exp} - y_{i,j}^{pred} \right| \tag{19}$$

$$RE = \frac{1}{N \times M} \sum_{j=1}^{M} \sum_{i=1}^{N} \left( \frac{y_{i,j}^{exp} - y_{i,j}^{pred}}{y_{i,j}^{exp}} \right) \tag{20}$$

## 5. Results and Discussion

### 5.1. ANN Model Training

The datasets were divided into three parts: 70% for training, 20% for validation, and 10% for testing, which contained 28, 8, and 4 sub-datasets, respectively. Figure 5 displays a comparison between the real data and the predicted result from the ANN model during the training mode. The MSE was $4.0519 \times 10^{-5}$, $5.0850 \times 10^{-5}$, and $8.9327 \times 10^{-6}$ for the weight fractions, temperatures, and pressures, respectively, while the average MSE for all data was $4.1085 \times 10^{-5}$. The values of the MSE for temperature and pressure were lower than for the weight fractions. This was a result of the weight of the objective functions for temperature and pressure being higher than the weight of the weight fractions because the normalized variables for temperature and pressure fell within the range 0–1 while the weight fractions fell within the range from 0–0.26. The very low level of the predicted errors confirms the reliability of the proposed ANN model.

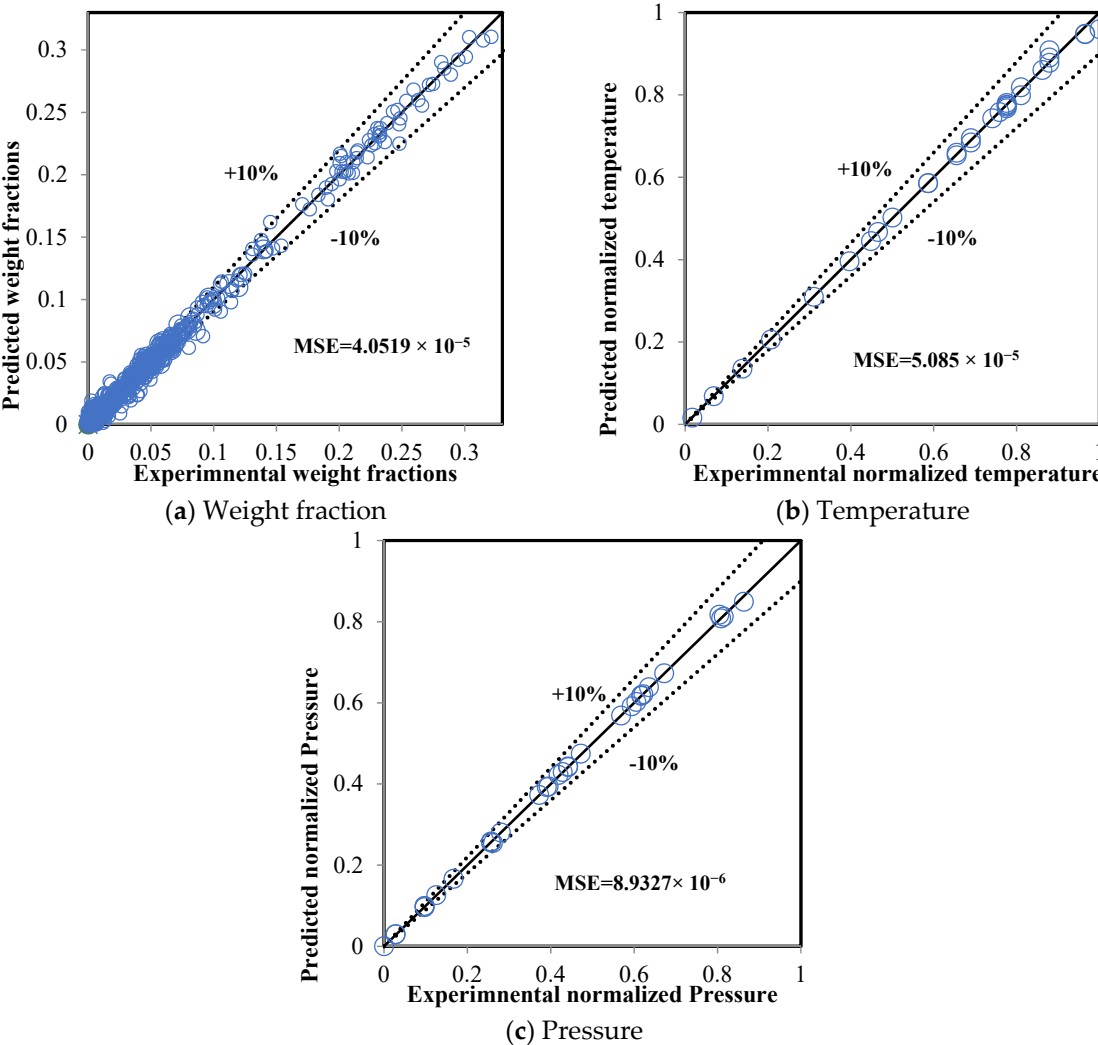

**Figure 5.** Comparison between the actual and ANN results (training mode).

### 5.2. Comparison between the ANN and RMM Predictions

The ANN model that was validated in the learning mode was applied to estimate the model outputs in the testing mode; four sub-datasets were used in compassion between the actual data and predicted results. The catalyst deactivation term was already present in the two models, which included the ANN model as time normalized from 0–1225 to 0–1 and also added the kinetic model to achieve the predictions of the RMM model. Figures 6–8 show a comparison between the actual and predicted results using the ANN and RMM, while the summary of the predicted errors of two models is illustrated in Table 6. Figure 6a,b represents a comparison between the actual and predicted weight fractions obtained by the RMM and ANN models, respectively. The corresponding values of the $R^2$, MAE, and MSE error functions achieved by the ANN model were 0.9403, 0.0062, and 0.002044, respectively, while for the RMM model, they were 0.9318, 0.0070, and 0.0002284, respectively. The MRE does not calculate for the weight fractions because some real values (those that were equal to zero) displayed an infinite relative error. Comparisons between the real and predicted outlet reactor temperatures obtained by the RMM and ANN models are presented in Figure 7a,b, respectively. The corresponding values of the $R^2$, MAE, MRE, and MSE error functions achieved by the ANN model were 0.9736, 2.2246, 0.4774 and 21.6587 respectively, while for the RMM model, they were 0.8951, 4.0939, 0.5679 and 37.329, respectively. Figure 8a,b shows a comparison between the actual and predicted outlet reactor pressures obtained by the RMM and ANN models, respectively. The corresponding

values of the $R^2$, MAE, MRE, and MSE error functions achieved by the ANN model were 0.9467, 0.4010, 1.4129, and 1.1988, respectively, while for the RMM model, they were 0.4859, 1.6248, 6.4959, and 5.303, respectively. For the RMM model, the lower value of $R^2$ in the case of pressure can be attributed to the fact that the error in pressure was accumulative from the first reactor to last one, in contrast to the temperature, whereas the heat exchangers were used to heat the feed to the required temperature before entering each one of these four reactors, so there would be no accumulative error. From these figures, close mappings between the measured and simulated weight fractions, temperatures, and pressures can be observed for the two models.

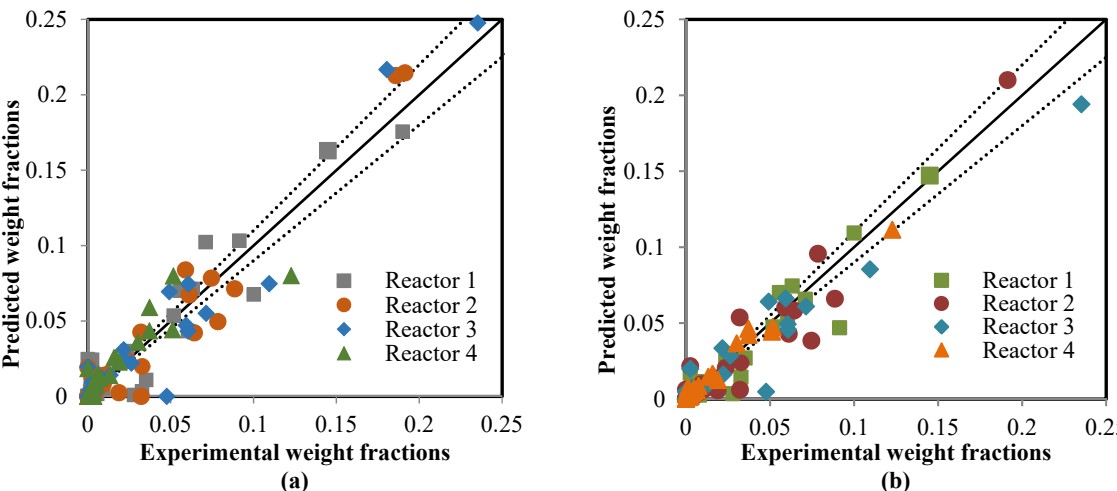

**Figure 6.** Comparison between the real and predicted weight fractions: (**a**) mathematical model, and (**b**) artificial neural network.

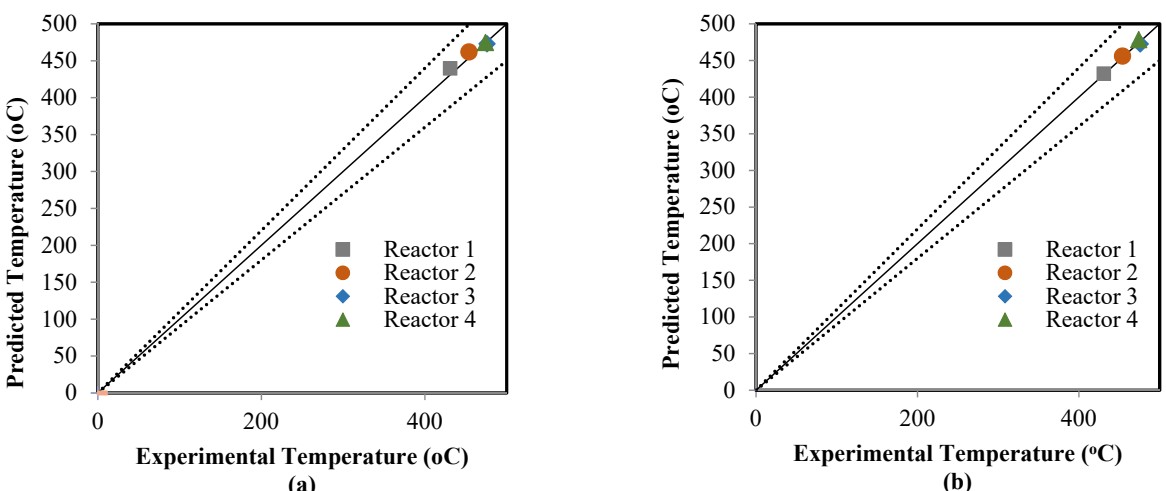

**Figure 7.** Comparison between the real and predicted temperatures: (**a**) mathematical model, and (**b**) artificial neural network.

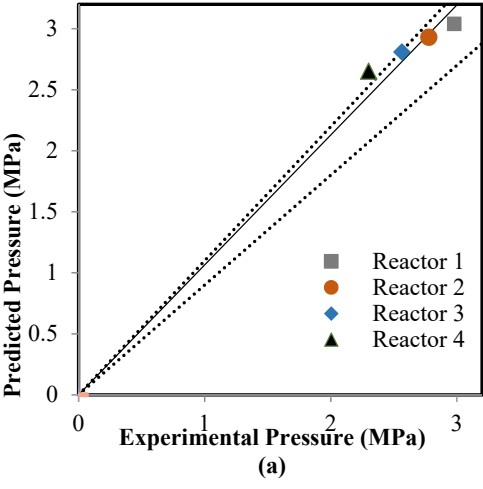
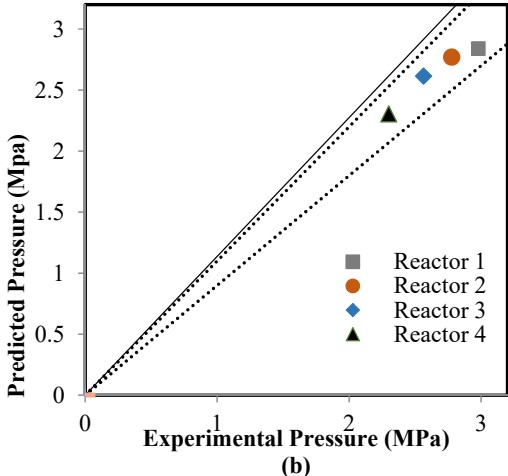

**Figure 8.** Comparison between the real and predicted pressure: (**a**) mathematical model, and (**b**) artificial neural network.

**Table 6.** Summary of the estimated errors.

| | **Mathematical Model** | | | **Artificial Neural Network** | | |
|---|---|---|---|---|---|---|
| | **Composition** | **Temperature** | **Pressure** | **Composition** | **Temperature** | **Pressure** |
| Coefficient of determination ($R^2$) | 0.9318 | 0.8951 | 0.4859 | 0.9403 | 0.9736 | 0.9467 |
| Mean absolute error (MAE) | 0.0070 | 4.0939 | 1.6248 | 0.0062 | 2.2246 | 0.4010 |
| %Mean relative error (MRE) | - | 0.5679 | 6.4959 | - | 0.4774 | 1.4129 |
| Mean square error (MSE) | $2.284 \times 10^{-4}$ | 37.329 | 5.303 | $2.0443 \times 10^{-4}$ | 21.6587 | 1.1988 |

Both the ANN and RMM models displayed notable errors because predictions made using both models depended on four consecutive calculations to predict the outputs of the four reactors, which led to the accumulation error increasing from the first reactor effluent to that of the last reactor. The training data for the ANN model represented only data for input into the reactor and output from the reactor. Therefore, the ANN model is valid for the data within this range. Using catalyst weights smaller than those used in training the ANN model makes the model display unusual values of the weight fractions.

The heavy naphtha reforming process is nonlinear, and there is a strong interaction between the products' compositions, temperatures, and pressures. The dehydrogenation of paraffins and naphthenes to aromatics entails endothermic reactions that occur quickly; they take place in the first and second reactors, causing a rapid temperature drop in these reactors. The temperature drop through the first two reactors was higher than that of the last two reactors due to the exothermic nature of hydrocracking and the dehydrocyclization reaction involving the paraffins that occurred in the first two reactors. Despite this process being nonlinear, these two models predicted the four reactors' effluent weight fractions, temperatures, and pressures very well, but the ANN model's predictions were slightly better than those of the RMM model. The ANN and RMM models can effectively simulate the complicated chemical reactors, but the ANN model is faster and more accurate than the RMM model, which is in agreement with the conclusions obtained by Elçiçek et al. (2014) [32]. The computational time was 18.3 s for the ANN model and 120 s for the RMM. This great difference in the computation time is due to the fact that solving the ANN model involves substituting input variables in simple algebraic equations to estimate the outputs of the process, while solving the RMM model involves solving 34 ordinary simultaneous

differential equations using the fourth-order Runge–Kutta integration method sequentially for each one of the four reactors. The validated ANN and RMM models can be used in the future for accurate simulation of industrial heavy naphtha reforming processes.

## 6. Conclusions

In the present study, real data obtained from the heavy naphtha reforming process in a long time period (1225 days) were modeled by both the rigorous mathematical model (RMM) and artificial neural network (ANN) model. The parameters of the ANN model, including the number of hidden layers, number of neurons in the hidden layers, and the number of iterations, were optimized to construct an optimal ANN model having 36-10-10-10-34 topology. To evaluate the goodness of fit, an error analysis was performed using the mean square error (MSE), coefficient of determination ($R^2$), mean relative error (MRE), and mean absolute error (MAE). The ANN model provided a precise and effective prediction of the experimental data with a coefficient of determination ($R^2$) of 0.9403, 0.9736, and 0.9467 for the weight fractions, temperatures, and pressures, respectively. In comparison, for the rigorous mathematical model, the coefficient of determination ($R^2$) was found to be 0.9318, 0.8951, and 0.4859 for the weight fractions, temperatures, and pressures, respectively. The $R^2$ of the ANN was higher than 0.94 for the weight fractions, temperatures, and pressures, indicating a good fit by the ANN for the testing dataset. All predictions of the error functions yielded lower values for the ANN model than the RMM model, suggesting that the ANN model is the most suitable model to describe the heavy naphtha reforming process. One disadvantage of the ANN model is that it cannot predict the process performance at the intermediate points inside reactors. We conclude that the ANN may be preferable as an alternative approach instead of the RMM to predict the performance of the heavy naphtha reforming process.

**Author Contributions:** Conceptualization, A.A.-S. and T.M.A.; methodology, A.A.-S.; software, Z.M.S.; validation, A.A.A., H.S.M. and Z.M.S.; formal analysis, Z.M.S.; investigation, A.A.-S.; resources, A.A.-S.; data curation, Z.M.S.; writing—original draft preparation, Z.M.S.; writing—review and editing, A.A.A.; visualization, T.M.A.; supervision, A.A.A.; project administration, A.A.A.; funding acquisition, H.S.M. All authors have read and agreed to the published version of the manuscript.

**Funding:** The authors have no relevant financial or nonfinancial interests to disclose.

**Data Availability Statement:** All relevant data are included in the paper.

**Acknowledgments:** The authors acknowledge the support of Department of Chemical Engineering, University of Technology Baghdad/Iraq, and Al-Mustaqbal University College, Hilla, Babylon, Iraq. Also, the authors acknowledge the collaboration of the staff of the testing laboratory in Middle Refineries Company for their efforts in providing the datasets used in this study.

**Conflicts of Interest:** We certify that they have no affiliations with or involvement in any organization or entity with any financial interest or nonfinancial interest in the subject matter or materials discussed in this manuscript.

## Nomenclature

| | |
|---|---|
| $A_i$, $B_i$, $C_i$, $D_i$ | Specific heat constants |
| Ac | Cross sectional area of the reactor ($m^2$) |
| a | Catalyst activity |
| $C_P$ | Specific heat (kJ/kmole.K) |
| $D_p$ | Catalyst particle diameter (m) |
| $E_A$ | Activation energy (J/mole) |
| $F_i$ | Molar flow rate of species i (kmole/hr) |
| G | Mass flux (kg/$m^2$ s) |
| $g_c$ | Acceleration of gravity (m/$s^2$) |
| $H_j$ | Molar enthalpy (kJ/kmol) |
| $H_f^o$ | Enthalpy of formation (kJ/kmol) |
| $k_i^o$ | Pre-exponential factor |

| | |
|---|---|
| $k_i$ | $i^{th}$ reaction rate (kmole $hr^{-1}$) |
| $k_d$ | Rate constant of catalyst deactivation ($day^{-1}$) |
| M | Dataset number |
| N | Component number |
| $P_i^o$ | Partial pressure of $i^{th}$ component (MPa) |
| $P_t$ | Total pressure (MPa) |
| R | Universal gas constant (J/mole K) |
| r | Rate of reaction (kmole/kgcat hr) |
| S | Stoichiometry of reaction |
| t | Time (day) |
| T | Temperature (K) |
| $T^o$ | Reference temperature (K) |
| w | Weight of catalyst (kg) |
| y | Weight fraction |
| Z | Reactor length (m) |
| **Superscript** | |
| norm | Normalized |
| **Subscript** | |
| exp | Experimental |
| i | Component number |
| j | Reaction number |
| min | Minimum |
| max | Maximum |
| pred | Predicted |
| **Greek letters** | |
| $\varepsilon_b$ | Void fraction ($m^3/m^3$) |
| $\alpha$ | Power of pressure effect |
| $\mu$ | Viscosity (kg/m·s) |
| $\rho$ | Density ($Kg/m^3$) |
| $\rho_c$ | Catalyst density ($Kg/m^3$) |
| $\Delta H_{Rj}$ | Heat of reaction (kJ/kmole) |

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
