# Peer review of "Comparison between Artificial Neural Network and Rigorous Mathematical Model in Simulation of Industrial Heavy Naphtha Reforming Process"

_catalysts, doi:10.3390/catal11091034_

Round 1
Reviewer 1 Report
In this work, the authors developed a ANN model to estimate the performance of heavy naphtha reforming process, and compared its accuracy with a deterministic mathematical model. The ANN turned out to be slight better than the RMM model in accuracy and computational efficiency, but failed to predict the performance in the internal points of reactors. Since artificial intelligence (AI) has great potential to replace traditional mathematical models in many chemical engineering processes, the findings in this work will be definitely interesting to the catalysis community. Therefore, I recommend its publication on Catalysis. The minor issues that the authors should pay attention are the following.
- The quality of the figures should be improved.
- The type of ANNs explored in this works is somewhat limited. Other types of ANN might have better performance in this particular task.
Author Response
Dear reviewer,
Thank you very much for effort on reviewing our paper. Fortunately, I have corrected the manuscript according to your useful comments. All the items which you mentioned are covered.
All the corrected items which you referred can be seen as red color text in the manuscript to facilitate the recheck process. Finally, we highly appreciate your notes which participate in upgrading our manuscript.
- The quality of the figures should be improved.
Response: improved.
- The type of ANNs explored in this works is somewhat limited. Other types of ANN might have better performance in this particular task.
Response: The authors would like to deep thank the reviewer for his/her valuable comments. Response to this comment “The ANN topology had minimum MSE when the number of hidden layers, number of neurons in the hidden layer and the number of training epochs were 3, 10 and 100000 respectively” please see the abstract.
Reviewer 2 Report
Dear Authors,
Before being ready for publication, there are some small things that have to be explained/formulated in a better way. Please find attached the comments/suggestions in no specific order:
- In the abstract section, the authors are mentioning that, in order to get 32 compositions, they solved their rigorous mathematical model consisting in 34 differential equations. Probably the other two variables were the temperature and pressure along the reactor, but this is not straightforward for the reader. Please be more specific.
- The authors are mentioning the topology of the ANN as (36-10-10-10-34). For the readers familiar with the ANN theory, these numbers are meaningful, but for the others they mean nothing. Please be more specific.
- In equation (9) appears the variable , but this variable meaning Is not present in the notations list. Please correct.
- As the model is based on pseudo-components, could the authors explain how they used the data taken from the book of Reid et al (1987) to estimate the properties of these components?
- In Table 3 there is a typing error for the measurement unit of the activation energy. Please recheck the typing errors throughout the manuscript as there are some more others. (Table 3: “Isomrization”, “Dehyhydrgenation”, etc)
- The authors are stating that “Neural networks are a series of mathematical algorithms that simulate the processes 151 of the human brain to estimate”. Most probably the ANN are mimicking the functioning of the human neural network. They do not simulate this behavior, as the subject of this paper is not related with the human anatomy. Please rephrase
- Lines 180-181: Equation should be written with small letter, the number of equations should be written as (15)
- The quality of Figures 2-4 should be improved.
- In the manuscript it is stated that “The most significant variable in the simulation results is the weight fractions of the 248 products because it is necessary to calculate the RON of the reformate.”. Maybe it will be useful for the readers to provide the methods for calculating RON number from the weight fractions.
- The authors should provide a justification for the low value of the coefficient of determination (R2) for the outlet pressures estimation using the RMM (0.485). The Ergun equation usually predict in an accurate way the pressure drop along the reactor.
- To compare the run times for the two approaches, details regarding the method of integration the differential equations should be provided (software, integration function, etc)
- The authors are mentioning “long time period (1255)” but the number does not have a measurement unit. Please be more specific.
Author Response
Dear reviewer,
Thank you very much for effort on reviewing our paper. Fortunately, I have corrected the manuscript according to your useful comments. All the items which you mentioned are covered.
All the corrected items which you referred can be seen as red color text in the manuscript to facilitate the recheck process. Finally, we highly appreciate your notes which participate in upgrading our manuscript.
- In the abstract section, the authors are mentioning that, in order to get 32 compositions, they solved their rigorous mathematical model consisting in 34 differential equations. Probably the other two variables were the temperature and pressure along the reactor, but this is not straightforward for the reader. Please be more specific.
Response: Done
- The authors are mentioning the topology of the ANN as (36-10-10-10-34). For the readers familiar with the ANN theory, these numbers are meaningful, but for the others they mean nothing. Please be more specific.
Response: The author would like to thank the reviewer very much for his/her comment. Response to this comment the abstract was modified and now it appears clearer.
- In equation (9) appears the variable, but this variable meaning Is not present in the notations list. Please correct.
Response: all the variables meaning in equation (9) were inserted in the notations list.
- As the model is based on pseudo-components, could the authors explain how they used the data taken from the book of Reid et al (1987) to estimate the properties of these components?
Response: The physical and thermodynamic properties of the pseudo-components was assumed to be equal to the properties of the main component for these pseudo-components. The physical properties were obtained from reference [26].
- In Table 3 there is a typing error for the measurement unit of the activation energy. Please recheck the typing errors throughout the manuscript as there are some more others. (Table 3: “Isomrization”, “Dehyhydrgenation”, etc)
Response: Done, please see Table 3.
- The authors are stating that “Neural networks are a series of mathematical algorithms that simulate the processes 151 of the human brain to estimate”. Most probably the ANN are mimicking the functioning of the human neural network. They do not simulate this behavior, as the subject of this paper is not related with the human anatomy. Please rephrase
Response: Done.
- Lines 180-181: Equation should be written with small letter; the number of equations should be written as (15)
Response: Done, for all equations
- The quality of Figures 2-4 should be improved.
Response: Improved.
- In the manuscript it is stated that “The most significant variable in the simulation results is the weight fractions of the 248 products because it is necessary to calculate the RON of the reformate.”. Maybe it will be useful for the readers to provide the methods for calculating RON number from the weight fractions.
Response: This paragraph was deleted from the manuscript because the we don’t have the experimental data to relate the RON with the molecular weights of the individual components.
- The authors should provide a justification for the low value of the coefficient of determination (R2) for the outlet pressures estimation using the RMM (0.485). The Ergun equation usually predict in an accurate way the pressure drop along the reactor.
Response: The authors would like to deep thank the reviewer for his/her valuable
comments. Response to this comment “ For RMM model, the lower value of R2 in the case of pressure is can be attributed to the fact that the error in pressure is accumulative from the first reactor to last one, in contrast to the temperature, where is the heat exchangers was used to heat the feed to the required temperature before entering each one of these four reactors, so there will be no accumulative error”.
- To compare the run times for the two approaches, details regarding the method of integration the differential equations should be provided (software, integration function, etc)
Response: This great difference in the computation time is due to that solving ANN model involve substitute input variables in simple algebraic equations to estimate the outputs of the process, while solving RMM model involve solving 34 ordinary simultaneous differential equations using fourth order Runge-Kutta integration method sequentially for each one of the four reactors
- The authors are mentioning “long time period (1255)” but the number does not have a measurement unit. Please be more specific.
Response: Done, 1255 days.